# Impact and characterization of serial structural variations across humans and great apes

Wolfram Höps[1], Tobias Rausch [1,2], Michael Jendrusch[1], Human Genome Structural Variation Consortium (HGSVC)*, Jan O. Korbel [1,3,24] & Fritz J. Sedlazeck [4,5,24]

Modern sequencing technology enables the systematic detection of complex structural variation (SV) across genomes. However, extensive DNA rearrangements arising through a series of mutations, a phenomenon we refer to as serial SV (sSV), remain underexplored, posing a challenge for SV discovery. Here, we present NAHRwhals (https://github.com/WHops/NAHRwhals), a method to infer repeat-mediated series of SVs in long-read genomic assemblies. Applying NAHRwhals to haplotype-resolved human genomes from 28 individuals reveals 37 sSV loci of various length and complexity. These sSVs explain otherwise cryptic variation in medically relevant regions such as the *TPSAB1* gene, 8p23.1, 22q11 and Sotos syndrome regions. Comparisons with great ape assemblies indicate that most human sSVs formed recently, after the human-ape split, and involved non-repeat-mediated processes in addition to non-allelic homologous recombination. NAHRwhals reliably discovers and characterizes sSVs at scale and independent of species, uncovering their genomic abundance and suggesting broader implications for disease.

Continuous advances in single-molecule sequencing technologies drive the discovery of increasingly complex patterns of genetic variation in the human genome, particularly in repetitive regions. These highly rearranged regions and the respective complex alleles contribute to population diversity and impact a wide range of phenotypes[1–4]. Investigating complex alleles in distinct population ancestries is important for elucidating their pathogenic impact, as well as their recent and past evolution. In spite of continued advances in genome assembly, we are far from understanding the full spectrum of human genetic variation particularly in repeat-rich genomic regions, leaving much of their impact on evolution, diversity, and human diseases in the dark[5–8].

Structural variation (SV) in the genomes of humans and other animals is increasingly characterized with complete allelic resolution, driven by advances in long read and homolog-preserving genomic technologies[9–11]. Recent long-read studies have revealed growing numbers of rearrangements of a complexity that escape analysis using conventional short-read sequencing. These rearrangement patterns are caused either by (i) complex genomic rearrangement processes[12,13], or (ii) by serial rearrangement events that accumulated at a locus gradually over evolutionary times, or rapidly due to its tendency to undergo SV[12,14]. The latter events are presumed to be particularly predominant in genomic regions harboring segmental duplications (SDs), which facilitate de novo rearrangements via non-allelic homologous recombination (NAHR)[12]. This spatial preference makes their discovery and interpretation especially challenging – with short read based mappers struggling in SD-rich regions – while raising the notion

[1]European Molecular Biology Laboratory, Genome Biology Unit, Meyerhofstr. 1, 69117 Heidelberg, Germany. [2]Molecular Medicine Partnership Unit, European Molecular Biology Laboratory, University of Heidelberg, Heidelberg, Germany. [3]European Molecular Biology Laboratory, European Bioinformatics Institute, Wellcome Genome Campus, Hinxton, Cambridge CB10 1SD, UK. [4]Human Genome Sequencing Center, Baylor College of Medicine, Houston, TX 77030, USA. [5]Department of Computer Science, Rice University, Houston, TX, USA. [24]These authors jointly supervised this work: Jan O. Korbel, Fritz J. Sedlazeck. *A list of authors and their affiliations appears at the end of the paper. ✉e-mail: jan.korbel@embl.de

that such SVs may be especially relevant among the fraction of human variation that remains to be discovered[15]. In light of their 'serial' nature, we refer to the latter as *serial structural variants* (*sSV*) for the purpose of this manuscript.

While the frequency of sSVs in healthy and diseased individuals is poorly explored, these regions are important for population and medical genetics—since they demarcate regions with high diversity in haplotype structure, and regions prone to undergo de novo rearrangements including pathogenic copy-number variation (CNV)[13,16–18]. As part of a comprehensive survey of polymorphic inversions in the human genome, we recently reported several isolated sSV-like events[14], which included rearrangements that likely facilitate - or protect against - disease-causing copy number variations in the human genomic loci 3q29, 15q13.3 and 7q11.23. In addition to this, a range of further studies reported medically relevant SD-associated CNVs that could be interpreted as sSVs, including mutational events causative for the Coffin-Siris syndrome, cone-rod dystrophy, intellectual disability and seizure and neonatal hypoxic-ischemic encephalopathy in patients[17]. Similarly, sSV events potentially causative for early-onset neuropsychiatric disorders[19] and Angelman syndrome[20] have been identified. From a population genetics viewpoint, the *TCAF1/2* locus displays substantial human-specific sSV-like copy-number variation associated with positive selection and an implicated role in adaptation[21] and the *TBC1D3* gene family, similarly, displays remarkable human diversity attributable to sSVs implicated in the expansion of the human prefrontal cortex[22]. Recently, the first pangenome-graph reference released by the Human Pangenome Reference Consortium (HPRC) provided new insights into highly variable complex regions, such as the *RHD*, *HLA-A*, *C4*, *CYP2D6* and *LPA*-containing loci, many of which harbor interspersed SDs and are thus likely candidates for sSV activity[23]. Indeed, for example, SD-associated variation in the LPA locus has been reported to directly impact the risk for cardiovascular diseases[24]. Yet, despite their undisputed relevance, these regions remain poorly resolved and consequently understudied, owing to challenges in the discovery and interpretation of genetic variation in these regions.

The identification of sSVs using long-read sequencing[11,25] or long-read based genome assemblies[9,26,27] is conceptually separate from 'classical' SV calling[28], as even a perfect description of sequence alterations (e.g., 'Del-Inv-Del') does not necessarily capture the series of underlying simple SVs (such as an inversion followed by a deletion, denoted 'Inv + Del' in Fig. 1A). Even when this 'mechanistic' viewpoint is ignored, only few methods for resolving complex patterns of SV have been developed, and these come with remaining limitations[11,29]. These limitations include the need for specialized algorithms designed to capture complex multi-breakpoint SVs thought to be formed through a single mutational event[11] dependent on the region of the genome these tools are applied to. De novo genome assemblies, which since recently achieve nearly the full human genome[30], theoretically allow for a more comprehensive study of sSV. However, the identification of sSV from assemblies remains challenging[31] and to date has not been explicitly addressed through computational methods. Thus, novel methodologies are required to address our current lack of detection and improve our understanding of how sSV contributes to genomic variation.

To directly address this existing gap, we devised a computational framework, NAHRwhals (<u>NAHR</u>-directed <u>W</u>orkflow for catc<u>H</u>ing seri<u>AL</u> <u>S</u>tructural Variations), which allows to infer regions likely to have undergone consecutive overlapping SVs (i.e., sSVs). Given a genomic assembly and genomic reference coordinates of interest, NAHRwhals identifies structural differences and NAHR-enabling repeats and employs an exhaustive search over potential sSVs that can explain the observed difference in sequence architectures—to generate hypotheses about serial rearrangements resulting in sSVs across the genome. NAHRwhals thereby leverages the sequence resolution of genome assemblies, to enable identification of patterns of complex variation otherwise inaccessible. The tool can be readily applied using any genome assembly and a freely exchangeable reference sequence, making it suitable for comparative genomic analysis. Applying NAHRwhals, we reveal the occurrence of sSVs in 28 diploid genome assemblies—5 of which were previously unpublished – and highlight implications in medically important regions.

## Results

### Automated detection of serial structural variations (sSV) from genome assemblies

We developed the NAHRwhals framework to allow systematic identification of sSVs in haplotype-resolved genome assemblies. NAHRwhals can be run in two primary modes: genotyping mode and whole-genome mode. The required inputs are:

(1) Reference Genome ('*Ref*'): a reference FASTA file, such as GRCh38.
(2) Query Genome ('*Query*'): A single-haplotype assembly FASTA file to be analyzed.
(3) Regions of Interest ('*ROI*'s) - Required only in genotyping mode: coordinates on the reference genome ('*Ref*') to be genotyped. In whole-genome mode, NAHRwhals automatically determines *ROI*s by conducting an initial alignment of the entire reference and query genomes to identify discordant regions, which are then used as *ROI*s ("Methods", Fig. S1).

NAHRwhals genotypes any ROI in four steps: (i) isolating the *ROI* in *Ref* and locating its counterpart in *Query*, (ii) aligning the *ROI*-sequences from *Ref* and *Query*, (iii) turning this pairwise alignment into a simplified, '*segmented*', representation which facilitates repeat discovery and SV simulation, and (iv) employing a depth-first search to find sSV candidates that can rearrange the segments in *Ref* to mimic their order in *Query* (see also Fig. 1B; "Methods"). Below, we describe these four steps in detail:

#### i) Sequence retrieval

The *ROI* is extracted from the reference genome (*Ref*), resulting in *ROI-Ref*. Minimap2[32] is then used to locate the corresponding, potentially SV-carrying, region in the query genome (Query), yielding *ROI-Query*.

#### ii) Highly accurate pairwise alignments

A pairwise alignment between *ROI-Ref* and *ROI-Query* is produced using a custom pipeline. This pipeline involves splitting *ROI-Query* into chunks of 1 kbp, aligning these chunks to *ROI-Ref* individually (allowing for multi-mappings), and subsequently re-joining them ("Methods", Fig. S2). This method significantly improves alignment quality in repeat-rich regions compared to the default Minimap2 settings ("Methods"). Alignments shorter than 1kbp (default) are then discarded, and alignment coordinates are rounded to the nearest multiple of a rounding factor (default: 1 kbp for ROI ≤ 500 kbp, 10 kbp for ROI > 500 kbp) to eliminate small alignment incongruencies.

#### iii) Alignment segmentation

A segmentation algorithm simplifies the alignment by identifying uninterrupted stretches of alignment, referred to as "segments," which range in size from 1 kbp to several hundred kbp ("Methods", Fig. S3). This segmented representation of the alignment retains all information of the alignment, while significantly simplifying the identification of NAHR-enabling repeat pairs and the exhaustive search for sSVs in the subsequent step.

#### iv) Exhaustive search for sSVs

The segmented alignment, represented as a matrix, serves as the foundation for an exhaustive search for NAHR-based sSVs. This search employs a breadth-first search approach to explore possible

trajectories of sSVs, with the goal of transforming the reference genome segments (*ROI-Ref*, represented on the x-axis) into the corresponding query genome structure (*ROI-Query*) (Fig. 1C). At each stage of the search, the algorithm:

(1) generates a list of potential downstream NAHR-based SVs (deletions and duplications between pairs of segments in the same orientation, and inversions between pairs in inverse orientation).

(2) Simulates each SV by deleting, duplicating, or inverting respective columns of the alignment matrix, generating new pairwise alignment matrices.

(3) Calculates a 'segmented alignment score' which quantifies the percentage of correctly aligned segments between *ROI-Ref* and *ROI-Query*, scaled by segment size (Methods).

The algorithm explores this space up to a predefined maximum depth [default: 3], using a tree structure where each node represents a state of the genomic locus after applying previous SVs. Then, sSVs with a segmented alignment score within 5% of the best scoring one are reported. For downstream analyses, a simulation is considered 'successful' if it achieves an alignment score above a given threshold [default: 98%]. As default heuristics, we limit the maximum allowed

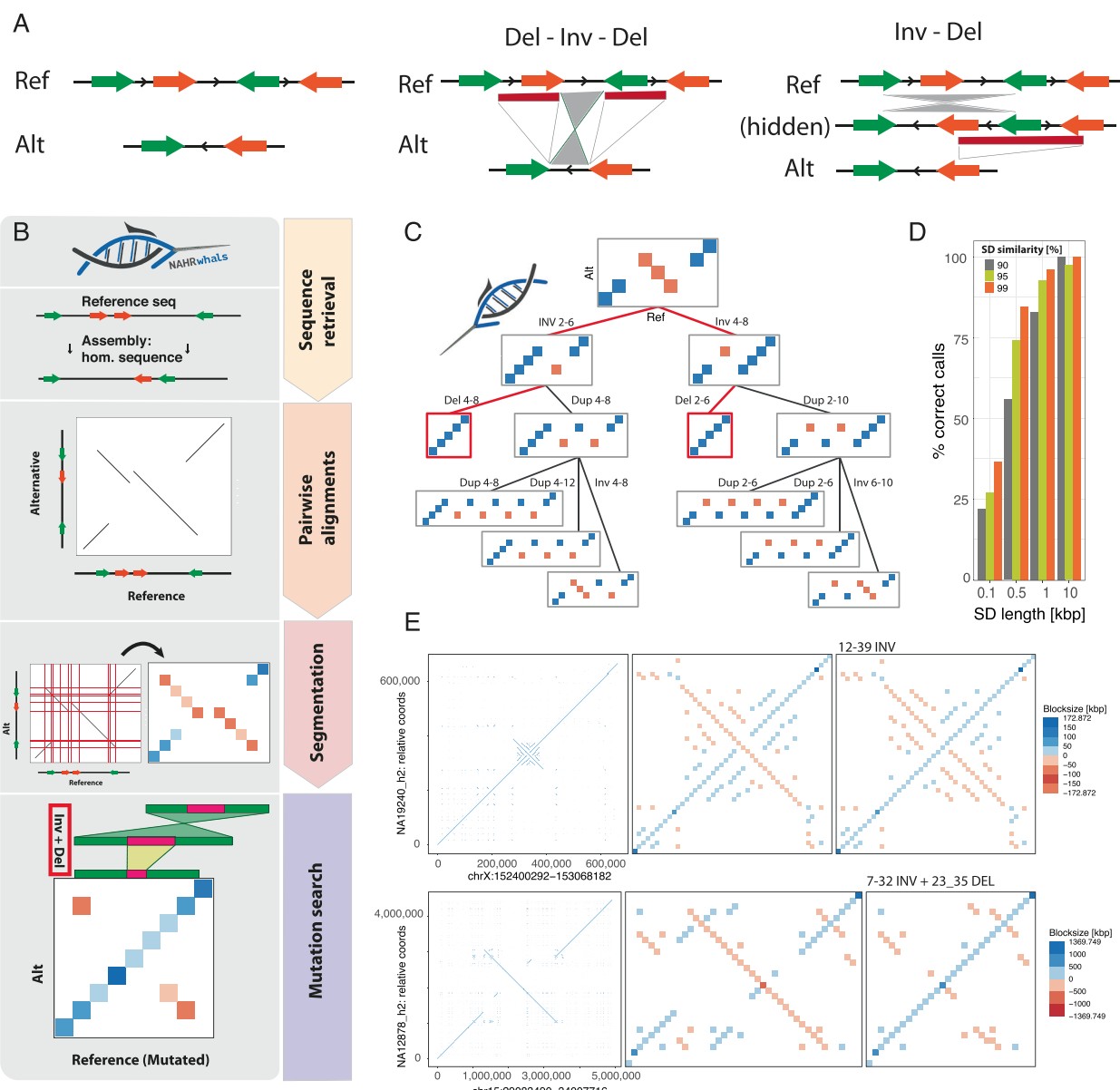

**Fig. 1 | Overview: the NAHRwhals sSV detection method. A** Schematic representation of a sequence pair illustrating the principle of serial SVs (sSVs). Traditional SV calling is often limited to detangling only simple SV (Del vs. Inv) or to report the entire allele (Del-Inv-Del; middle). Instead, NAHRwhals infers a series of simple SV which can explain a given structural haplotype outcome (Inv-Del; right). **B** Flowchart showing the key steps of the NAHRwhals algorithm. Starting from reference and alternative assemblies and a reference region of interest, the homologous region is first extracted from the assembly. Pairwise alignments between Ref and Alt are generated and transformed into a segmented representation. Using this segmented dotplot, an exhaustive search is invoked to explore possible series of NAHR-mediated rearrangements explaining the structural differences. **C** An example mutation search tree of depth 3 for a simple segmented dotplot. Successful serial SVs are highlighted in red. **D** Results of sSV calling on simulated runs. SD length and similarity correlate with prediction accuracy as longer/more similar sequences are more likely to be retained both in the initial alignment and the dotplot segmentation. **E** Two examples of segmentation and mutation calling in real loci. Blue: positive alignments, Red: reverse complement alignments. Shade: segment length. Left: dotplot of pairwise alignments between hg38 (x) and assemblies (y). Middle: segmented dotplot representation. Right: segmented dotplot after application of the highest-scoring series of SVs.

number of duplications per sSV to avoid 'exploding' sequence length [default: 2] and, per layer, retain only the X best-scoring nodes [default: Inf].

**Benchmarking of NAHRwhals.** To assess the performance of NAHR-whals, we generated and surveyed 1,200 artificial sequences containing two pairs of segmental duplications each, and simulated sSVs of depth one to three (e.g. Dup-Inv-Del) ("Methods", Fig. S4). We additionally randomized the length (0.1 kbp−10 kbp) and similarity (90% −99%) of segmental duplications to estimate thresholds above which pairwise alignment are typically reported and represented in the segmented dotplots. We find a positive correlation between genotyping accuracy and the length and similarity of repeats, with near-perfect sSV-detection accuracy if the repeat length exceeded 10 kbp in this simulated setting, while short (100's bp) or highly dissimilar repeats (<90%) are typically rejected from sSV simulation (Fig. 1D). To further confirm the viability of NAHRwhals in a more realistic setting, we applied NAHRwhals to ten previously identified inversion loci of varying complexity and known to be subject to NAHR[14]. In this benchmarking exercise, NAHRwhals obtained accurate segmented representations and expected SV genotypes for all ten loci (Fig. 1E, S5). We next tested NAHRwhals in whole-genome mode on the hg38 vs T2T reference to replicate the set of validated SVs published in ref. 33 (Fig. S3). Using all 'validated' SVs above 10kbp and a 25% reciprocal overlap criterion, out of 268 SVs >10kbp in the joint callset, 129 (48%) were exclusive to Yang et al., 84 (31%) exclusive to NAHRwhals and 55 (20%) jointly called. Among the 55 joint calls, genotypes differed in eight cases, in six of which we find NAHRwhals to be either equally ($N = 1$) or more ($N = 5$) accurate (Fig. S6). Yang et al.−exclusive calls were 1.3-times less frequently overlapping segmental duplications, suggesting that the SVs missed by NAHRwhals are enriched for non-NAHR events. Notably, all three largest SVs were deemed sSVs and were exclusive to NAHRwhals, although two regions appeared in non-validated callsets and without correct genotype. To demonstrate that NAHRwhals works independently of species, we also analyzed the Heinz1706 vs M82 Tomato plant assemblies[34], identifying four deletions explicable by NAHR (Fig. S7). In conclusion, our method shows promising results across simulated and previously characterized NAHR-affected regions along the genome.

**Automated reconstruction of sSVs across 336 loci along the human genome.** Having established the general ability of NAHRwhals to infer sSVs through simulations and example test loci, we next performed a broad survey for sSV events in 28 individuals (yielding 56 haplotype-resolved assemblies). These assemblies were generated by the Human Genome Structural Variation Consortium (HGSVC) using Pacific Biosciences long read sequencing. These include hitherto unpublished assemblies of 5 human genomes, amounting to 10 assembled haplotypes, generated using 27.7–47.2X coverage Pacific Biosciences long reads (HiFi) that were phased into chromosome-scale haplotypes using Strand-seq[35] (Methods). Since these 56 assemblies span human individuals of diverse population ancestry, they allow us to obtain an estimate for the prevalence of sSVs in humans and identify classes of sSV-mediated variation. To this end, we defined a list of potentially sSV-carrying genomic regions by applying a merging strategy in which we integrated: (a) all SV regions longer than 10 kbp determined in a previous comprehensive SV survey of 64 human haplotypes[9] ($n = 915$); (b) sites of polymorphic human inversions[14] longer than 10 kbp ($n = 290$); and (c) segmental duplications obtained through the UCSC Table Browser[36]. Using this procedure, we determined 336 non-overlapping regions of interest (ROIs, median length: 170.8 kbp), which were subsequently scanned for sSV content. In each of these regions, we tested the human along with four great-ape assembled haplotypes (Methods) for (s)SVs with respect to the CHM13-T2T-v1.1 genome assembly[30]. ROIs were provided in GRCh38-

coordinates to NAHRwhals, which converted input coordinates to CHM13-T2T using a custom segment-liftover procedure based on minimap2 (Methods).

By screening across these samples and loci, NAHRwhals inferred 37 loci with at least one instance of overlapping SVs likely to have arisen in a serial manner (Supplementary Data 1). Adjacent (i.e. non-overlapping) SVs were not considered serial. Among the remaining 299 regions, 8 regions did not display a contiguous assembly in any sample and 21 regions could not be explained by NAHRwhals (despite being assembled in at least one sample), indicative of the involvement of non-NAHR mediated mechanisms. All other loci (270) displayed only zero-stage (Ref) or single-stage SVs (Inv, Del, Dup) (Fig. 2A, S8). The group of 37 sSVs was subsequently retained for further analysis (Fig. 2B, Supplementary Data 2). Across the 37 sSV regions in 56 human haplotypes, we identified 163 SVs of predicted depth 2 or 3. SVs of depth 2 and 3 were generally rare, displaying average allele frequencies 2.3-fold and 4.8-fold lower than depth-1 SVs, respectively (Figs. S9, S10). Notably, 65% of all predicted intermediate states (e.g., 'Inv' for an 'Inv+Del' haplotype) were indeed observed in another sample (Fig. 2C), suggesting that most of these complex NAHR loci are the result of accumulation of serial, temporally distinct events. Furthermore, all 20/ 20 sSVs for which the predicted intermediate state was missing were rare events (allele count <3), suggesting that their intermediate states may be missing due to our limited sample size.

Reflective of the often complex nature of the loci, out of all 2109 sequences (corresponding to 37 loci in 56 de novo assemblies plus hg38), assembly breaks prevented detailed analyses of potential sSVs in almost one third (625/2109 (29.6%)). As expected, the number of contiguously assembled regions was correlated with assembly N50 (Fig. S11). Likewise, the length and SD-content of regions were negatively correlated with the rate at which they could be resolved (Fig. S12). Samples with self-identified ancestries from AFR showed a slight but significant enrichment in depth-1 and depth-2 SVs compared to samples of admixed american (AMR) and east asian (EAS) ancestry, respectively (Fig. S13). However, the low number of samples per superpopulation (AFR: 15, AMR: 3, EUR: 4, EAS: 5, SAS: 1) discouraged further ancestry-based analyses.

Lastly, 540/2109 sequences (25.6%) were considered as 'unexplained' as NAHRwhals did not indicate either a reference state or any NAHR events. We investigated these further by manually curating 35 of these alignments - one for each sSV loci which had at least one unexplained sample (Fig. S14). The majority indeed displayed either small-scale (12/35; 34.3%) or large-scale (7/35; 20%) non-NAHR rearrangements. Further 10 regions (28.6%) exceeded the borders of our window or no homologous alignment was found. In six cases (17.1%), NAHR-whals was too conservative in rejecting alignments for exceeding boundaries ($n = 4$) or missed an optimal solution due to prematurely aborting branches of the mutation search tree ($n = 2$).

To better visualize and report the sSVs results across multiple samples, we devised a visualization resembling a directed flowchart, in which each temporally distinct NAHR mutation event is represented by a node (Fig. 2D, F). To illustrate the types of sSVs identified with NAHRwhals, we first highlight a ~1.5 Mbp region on chromosomal region 1p11.2-1p12, which displays several pairs of overlapping SDs in the reference state (carried by CHM13-T2T, hg38 and four other assemblies). A simple inversion between one of the SD pairs was observed in 17 samples, and finally 15 samples were carrying a third configuration which presumably features deletion of the inverted haplotype, corresponding to an 'Inv + Del' sSV (Fig. 2D, E). Another example of a more complex sSV-rich region was found in chromosomal region 11p15.4. In this case, several distinct haplotype configurations were observed, which could be explained by one, two and three consecutive SVs, respectively (Fig. 2F, G). Our callset includes also other regions for which sSV-like patterns have been described previously, such as variation in regions containing *TCAF1/TCAF2* (7q35

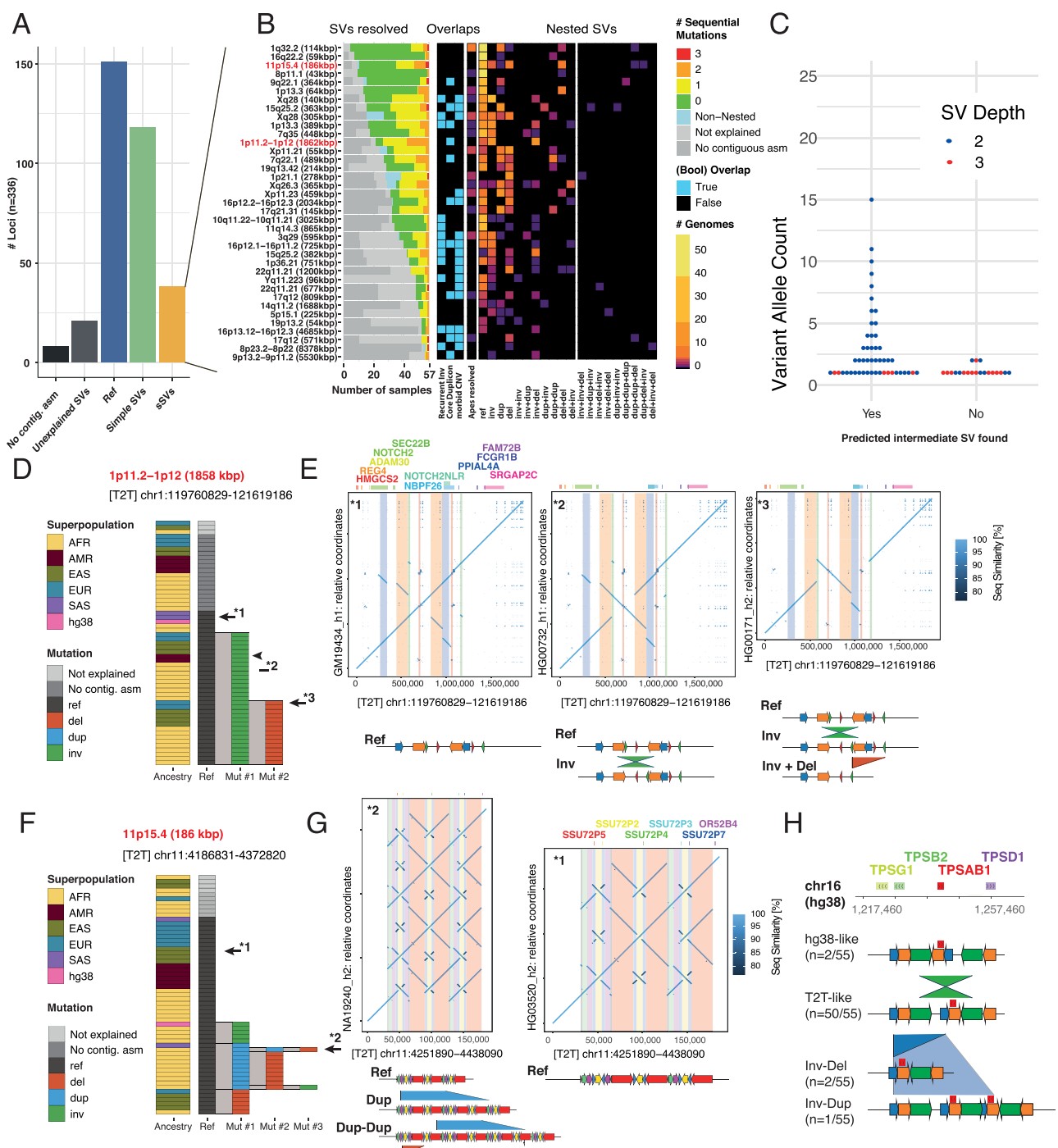

**Fig. 2 | Inversion regions identified as sSVs. A** Broad classification of the 336 loci initially surveyed with NAHRwhals. Loci were considered as containing sSVs if they displayed at least one overlapping pair of SVs in at least one sample. **B** Overview over the full callset of 37 inversion-containing loci in which sSVs were discovered in at least one sample. The diagram shows the prediction performance in humans and apes ('SVs resolved'), the presence of recurrent inversions, core duplicon-mapping genes and morbid CNV regions in the genomic region, as well as genotypes for each locus. **C** A beeswarm plot showing whether predicted intermediate SVs (e.g., 'inv' for an 'inv+del') have been found, as a function of the frequency of the sSV. **D** Three

distinct sequence configurations observed in the 1p11.2-1p12 sSV. 15/38 samples harbor a deletion preceded by an inversion compared to CHM13-T2T. **E** Dotplots and SD schematics illustrating examples of all three configurations. **F** A 185 kbp region on 11p15.4 showing complex patterns of nested SVs leading to extreme diversity in the region explicable by NAHR. **G** Dotplot and SD schematics of a highly rearranged (left) and a reference-like (right) alternative configuration. **H** sSVs found in the disease-relevant TPSAB1-containing region. Observed CNVs can be explained as simple SVs on the CHM13-T2T-like configuration, but appear as sSVs with respect to hg38 (Fig. S25).

(448 kbp))[21], *POTEM/POTEG* (14q11.2 (1688 kbp))[37], *TBC1D3* (17q12 (571 kbp))[22], *AMY1A-C* (1p21.1 (278 kbp))[10] and others (see Supplementary Data 1 for a full list)—all of which underwent dynamic SD-associated rearrangements in human and great ape evolution[21]. sSV plots and

dotplot visualization for each of the 37 sSVs are in the supplementary material (Figs. S15–S22).

In order to validate the inversion status of sSVs, we made use of an orthogonal sequencing technology, template strand sequencing

(*Strand-Seq*)[35], with data available for 19/28 samples. 11/37 regions exhibited sufficient unique sequence content to be examined with the technology[14]. Diploid inversion genotypes between Strand-Seq and NAHRwhals agree in 95/104 SVs (91%, Supplementary Data 1). Apart from inversions, Strand-Seq was able to resolve depth-2 CNVs in two loci, confirming all predicted SVs in these regions (Fig. S23, S24, "Methods"). We also compared our callset to rearrangements published by the HPRC[10], noting that 17/37 locations are exclusive to NAHRwhals and may have been missed in the HPRC dataset (10% reciprocal overlap). The sequences of the 37 sSV loci were additionally validated by mapping ultra-long nanopore reads (generated by the HGSVC, and available for 11/26 samples) directly to the respective assemblies and searching for homozygous variants using Sniffles2[38]. Alignment misassemblies should not be supported by any ONT reads, and thus appear as homozygous SVs ("Methods"). 45/796 sequences (5.7%) display homozygous indels above 10kbp which may point at collapsed or duplicated regions. Among screened haplotype configurations which were predicted by NAHRwhals only once, the ratio of indel-containing sequences was only marginally higher (2/19 sequences (10.5%)), suggesting that most reported calls are biologically meaningful.

We also re-analysed all 119 'simple SV' calls, this time using hg38 as a reference to account for the possibility that certain loci may appear as sSVs only in the context of a different reference allele. This analysis yielded four additional hg38-specific sSVs (Supplementary Data 1), three of which displayed 'Dup-Dup' alleles indicative of a shorter (or collapsed) allele represented in hg38. In the last identified region, a 49 kbp region on 16p13.3, hg38 represents a minor inverted allele which is predicted to undergo 'Inv-Del' ($n = 2$) and 'Inv-Dup' ($n = 1$) in three samples, while the same alleles appear as simple SVs when compared to CHM13-T2T (Fig. 2H). Notably, the region contains the *TPSAB1/TPSAB2* genes (Fig. S25), CNVs of which have been associated with Alpha Tryptasemia, a non-lethal hereditary disease affecting up to 4-6% of the population[39]. To our knowledge, the mechanistic background of these CNVs has not been clarified previously. Lastly, we note that, when mapped back to hg38, 7/37 (18.9%) sSV loci display long (>1 kbp) stretches of hard-masked bases, which prohibit faithful SV reconstruction, highlighting the importance of using a contiguous reference for studying sSV loci. Therefore, our screening reveals a high prevalence of repeat-rich regions for sSV formation and further indicates that complex variation in many dynamic human loci can be explained in the framework of sSVs.

**Population-based validation of the 'serial' nature of SVs.** We next explored the genetic relationship of sSV carrying haplotypes to test whether sSVs have formed in a multi-step process rather than through 'one-time' complex rearrangements. We hypothesized that, should the sSV hypothesis be true, haplotypes of higher sSV depth should be genetically more similar to their intermediate predecessor state rather than reference (i.e., 'inv-del' haplotypes should locally be more similar to 'inv' than to 'ref' haplotypes). To test this, we combined NAHRwhals-based genotypes with an external high-quality 1000 Genomes SNP callset[40]. A workflow based on WhatsHap[41] was used to synchronize haplotype assignments (h1/h2) in the two callsets (Methods).

In line with our expectations, we found a case of two 'dup-dup' carriers co-clustering with their putative 'dup' predecessor (Fig. S26) and a more complex case of a depth-three SV, in which carriers of ref or depth-1 SV co-cluster, while deeper SV states, 'dup+inv' and 'dup+inv +inv' form a separate haplotype cluster (Fig. S27). In the remaining cases, clustering patterns are less clear, with carriers of SVs of different 'depths' often highly intermixed and thus pointing towards multiple, separately emerging instances of the mutations (Figs. S28, S29). Such complex patterns, previously described in more detail e.g. for the TCAF locus[21], suggest that reconstruction of the ancestry of loci requires more in-depth analyses. The non-trivial inheritance patterns

are reminiscent of recurrent inversions[14], which are strongly enriched in sSV regions and which may explain part of to the complexity.

**sSV occurrence and SV complexity across Hominidae.** Having identified abundant sSV loci in humans, we next set out to examine to which degree sSVs in these regions are human-specific, as a prerequisite to gaining an understanding as to how these regions may have emerged evolutionarily. Repeat-associated variation is known to have contributed substantially to the evolution of modern humans[42], and hundreds of genomic regions display SD-mediated inversions between non-human primates and humans[43–45], frequently accompanied by secondary CNVs near their breakpoints[45]. We considered that a fraction of our human sSV loci, too, may have undergone substantial restructuring during great ape evolution, which may be explicable through sSVs. To test this, we turned our attention to sSVs in the four great ape genome assemblies[22] included in our dataset (Methods). To account for the overall higher sequence divergence, we chose a lower threshold parameter of 95% for considering sequence reconstruction successful. When reviewing the set of 37 CHM13-T2T-based human sSV loci, we find that in 16/37 loci (43.2%), NAHRwhals could determine rearrangements translating from the human locus configuration to that of at least one great ape variant (Fig. 3A).

Out of 37 loci, only four suffered from a lack of locus-spanning contigs in any of the ape assemblies, suggesting that the high ratio of unexplained variants (21/37 regions) does not stem primarily from a lack of assembled sequence quality, but may instead be attributed to non-NAHR-associated SVs or missed calls by NAHRwhals. To decide which of these is the case, we examined dotplot visualizations of these regions (Figs. S30–S32). Indeed, the unexplained loci consistently displayed advanced levels of rearrangements, frequently featuring large insertions, deletions and translocations, unattributable to NAHR, which are likely the result of other formation mechanisms, including duplicative transposition events[46] not modeled by our framework. We illustrate the scope of complexity of sSV loci in great apes along a 12 Mbp region on the p-arm of chr16 (Fig. 3B), which harbors two human sSV loci, neither of which could be explained in any great ape assembly. A dotplot visualization of these regions in the two contiguous ape assemblies (Bonobo and Orangutan) reveals that both sSV loci are part of larger, highly complex rearrangements that exceed the scale of the human sSV both in size and complexity.

In line with greater evolutionary distances involved, we notice an approximately twofold enriched fraction of 2- and 3-step vs 1- step SVs in apes compared to humans (simpleSVs/multi-stepSVs/fraction: humans: 302/163/1.85, apes: 11/12/0.92). We again highlight two examples of sSV loci here. First, the aforementioned *TCAF1/2*-containing region on 7q35 displays a set of overlapping SDs in the CHM13-T2T-configuration, which can transition into an "Inv+Del" state in Bonobo and Chimpanzee (Fig. 3C). The further distant species Gorilla and Orangutan display a somewhat analogous configuration, but also harbor additional insertions that cannot be explained by NAHR alone. We also find instances of more isolated non-NAHR events, such as a duplicated section on Xq28 containing cancer/testis antigen 1 (*CTAG1A/CTAG1B*) genes, which are implicated with a variety of cancers[47]. In our dataset, the CHM13-T2T assembly, Bonobo and Chimpanzee share the same locus configuration, which is distinct from the non-duplicated region in Orangutan. NAHR was again not sufficient to explain the implicated duplication event. Across these examples and the remaining dataset, our results suggest that in the majority of human sSV loci, NAHR alone is insufficient to explain inter-species variation, where consequently also other DNA rearrangement mechanisms such as MMBIR/FoSTes[48] or duplicative transposition[46,49] are likely at play.

**sSV regions co-locate with disease-causing CNVs, core duplicon genes and recurrent inversions.** NAHR-mediated recurrent

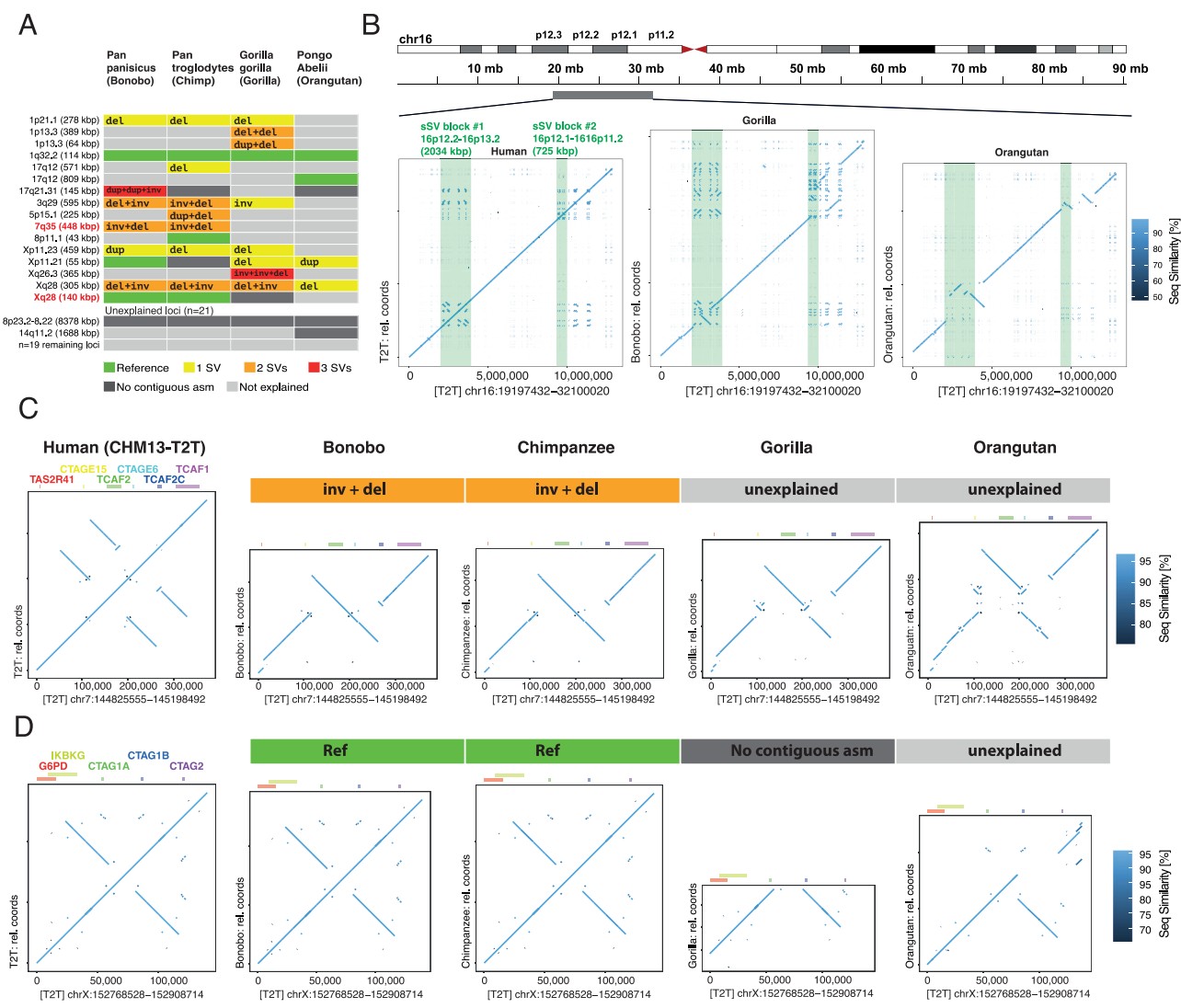

**Fig. 3 | Human sSV loci in great ape genome assemblies. A** Tabular view of NAHRwhals-based SV genotypes across 37 human sSV sites in four ape species. In 16 loci, the observed variation could be explained for at least one ape haplotype. **B** Dotplot views of a ca. 12 Mbp region on 16p, comparing the assemblies of CHM13-T2T, Bonobo and Orangutan (all single-contig; Chimpanzee and Gorilla: no contiguous asm). Two human sSV loci are highlighted in green. **C** Variation observed in the TCAF1/2 containing region on 7q35. Gorilla and Orangutan likely display a mixture of NAHR and non-NAHR SVs. **D** View of the CTAG1A/CTAG1B containing locus. Compared to Orangutan the CHM13-T2T-, Bonobo and Chimpanzee sequences harbor a duplication, which cannot be explained by NAHR.

inversions co-cluster with disease-causing CNVs in some of the most dynamically evolving regions of the human genome[14]. To anticipate whether our sSVs may be able to explain some of the variation in these regions, we initially tested sSV regions for spatial co-location with recurrent inversions and disease-associated CNVs (morbid CNVs) (Supplementary Data 3). In line with the tendency of (NAHR-promoting) SDs to flank morbid CNV sites[50–52], 16/37 sSVs (43.2%) overlapped with a morbid CNV region or its close surrounding (plus/minus 25% of the CNV length), compared to 51/299 (17.1%) of non-sSV loci, corresponding to a 3.7-fold enrichment of sSVs among morbid CNV-overlapping loci ($p = 4.9 \times 10^{-4}$, one-sided Fisher's Exact Test). To account for the possibility that this effect may be driven by locus size (i.e., larger loci may be more likely to exhibit sSVs while also being more likely to overlap morbid CNVs), we repeated the experiment, this time measuring if the midpoint of a locus falls into a morbid CNV. This did not alter the overall trend, with 15/37 midpoints of sSV-midpoints and 47/299 non-sSV-midpoints overlapping morbid CNV, respectively (odds ratio 3.63, $p = 7.1 \times 10^{-4}$, one-sided Fisher's Exact Test). Furthermore, 11/37 (29.7%) of sSV loci overlapped at least one member of a gene family mapping to core duplicons such as *GOLGA* and *NPIP*

("Methods", Supplementary Data 3), corresponding to a 6.6-fold enrichment compared to non-sSV loci where this was the case for 18/299 (6.0%) regions ($p = 5.6 \times 10^{-5}$, one-sided Fisher's exact test). This enrichment is in line with the role of core duplicons which are implicated in the expansion of segmental duplication and further repeat-driven mutational processes[53]. Lastly, sSVs were 6.25-fold more likely to overlap with recurrent inversions than non-sSV loci (12/37 (32.4%) vs 21/299 (7.0%) overlaps; $p = 3.78 \times 10^{-5}$, one-sided Fisher's exact test), supporting the notion that recurrent inversions are disproportionately associated with complex variation[14]. When again only sSV-midpoints were considered, the enrichment dropped to 3.66-fold (6/37 vs 15/299 overlaps, $p = 0.019$, one-sided Fisher's Exact Test). Consequently, our screening suggests that, among the 336 initially included loci, sSVs are strongly (3.66 to 6.6-fold) enriched in regions containing recurrent inversions, morbid CNVs and expanding SDs.

Despite our sample set consisting of phenotypically healthy individuals unlikely to carry morbid CNVs, we hypothesized that the sSVs focused on in this study may still follow similar mutational patterns as their morbid counterparts, and may help in uncovering mutational 'paths' leading to disease. We thus proceeded to explore

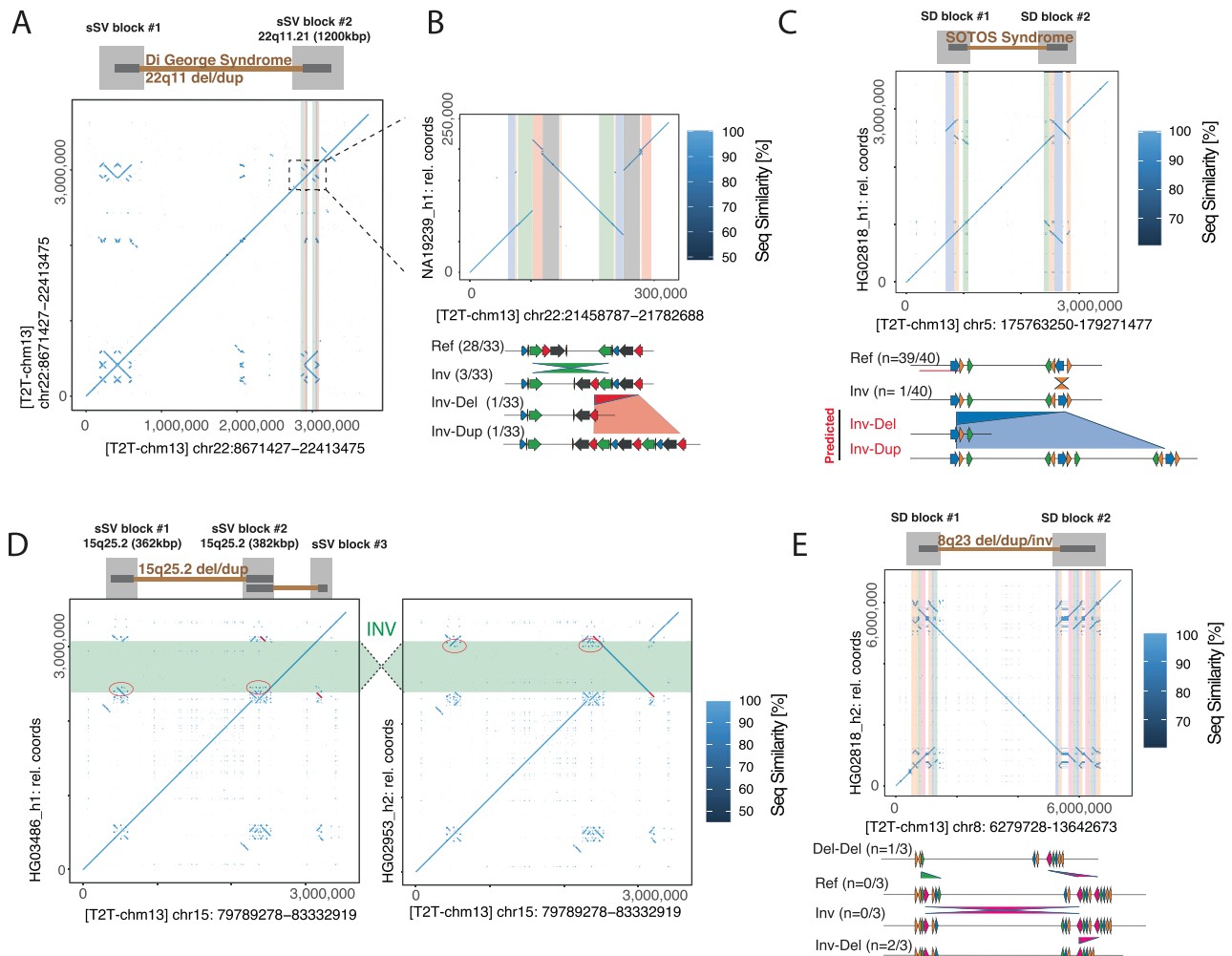

**Fig. 4 | sSVs in disease-relevant regions. A** Dotplot view of the 22q11 Dup/Del region, with two SD-rich blocks of sSV activity highlighted above. **B** Nested SVs lead to duplication or deletion of segmental duplications bordering the 22q11 Dup/Del region; potentially affecting the risk of subsequent CNV formation. **C** Inversion of one breakpoint of the SOTOS deletion creates a long pair of directly oriented SVs likely predisposing to subsequent CNV formation. **D** Two sSVs map to two ends of

the 15q25.2 deletion region, making both breakpoints susceptible to individual rearrangements. An inversion in one sample leads to transfer of SD sequence to a third sSV block. **E** Variation observed in the highly recurrent 8p23.1 Inv/Del/Dup region. Among the three resolved instances of this locus, NAHR-based variation was observed in both flanking SD blocks.

the molecular underpinnings of the co-location of inversions, morbid CNVs and sSV loci in our data. Indeed, by taking into account a larger window around morbid CNV co-locating sSVs, we found several instances of variants which we would consider likely 'premutative' under the sSV hypothesis. The first case discovered this way lies in chromosomal region 22q11, which can harbor local duplications and deletions (the latter being associated to DiGeorge Syndrome)[54]. The region contains a network of segmental duplications which are highly variable across humans and thought to represent mediators of the 22q11.2 deletion syndrome[55] (Fig. 4A). One sSV locus maps to a 1,200 kbp block of SDs flanking the region. The sSV features two pairs of overlapping pairs of inversely oriented SDs, out of which three divergent haplotypes emerge (Fig. 4B), two of which lead to expansion or contraction of the SD content, possibly increasing or decreasing the risk for subsequent formation of the larger disease-causing CNV. Our validation of this region with ONT reads (Methods) was not entirely conclusive, suggesting that further variants may be present in this region (Fig. S33A). However, the presence of three instances of the predicted intermediate (inverted) state in other samples (Fig. 4B) does support the emergence of 'Inv-Del' and 'Inv-Dup' haplotypes.

Primed by this initial example, we set up a more systematic screening to identify sSV events co-locating with morbid CNVs. To this end, we investigated a set of 113 disease-associated CNV locations[50,56], which we again subjected to a merging procedure to unify redundant calls and include surrounding SDs (Methods), leading to 48 non-redundant morbid CNV locations of interest which were tested with NAHRwhals across all 56 haplotype assemblies (Supplementary Data 4). Using this procedure, 35/48 loci displayed only reference states, non-contiguous assemblies or unexplained variance. Nine loci displayed single SVs, and another four displayed sSVs of depth 2 (Fig. S34, Supplementary Data 4). Focussing initially on SVs of depth 1, we report an inversion that can, under the sSV hypothesis, be interpreted as potentially premutative for formation of a ~ 2 Mbp CNV found in patients with the SOTOS syndrome[57]. Repeats at the flanks of this region have been noted before as substrates for NAHR[57,58]. Our inspection reveals a novel 258 kbp inversion of parts of the distal segmental duplication, present in 1/40 resolved haplotypes (2.5%) (Fig. 4C) and confirmed by orthogonal ONT-based validation. This inversion flips a substantial SD-containing segment (denoted as DLCR-A in previous literature[58]) into direct orientation with its proximal

counterpart, leading us to hypothesize that the inverted configuration could be at risk for subsequent morbid CNV formation.

Turning our attention to the four loci displaying sSV activity (regions 16p12.2-16p11.2 (12.4 Mbp), 15q25.3-15q25.2 (3.7 Mbp), 8p23.1-8p22 (6.9 Mbp), 3q29 (2.6 Mbp); Supplementary Data 4), we identified further regions in which the concept of sSVs yields hypotheses of CNV formation events. Since we described variation in the 3q29 region before, we here did not further focus on this region[14]. In the case of 15q25.3-15q25.2, we observed a ~700 kbp inversion which contains further SD fragments and effectively acts as a transporter of those SDs between two SD blocks (red circles, Fig. 4D). We also observed sSV activity in the 8p23.1 Inv/Del/Dup region in three resolved samples. Variation in this case is found in both flanking SD blocks (Del-Del) as well as between the blocks (Inv-Del) (Fig. 4E). All assembly configurations described here could be validated with ONT reads (Fig. S-33B-D). Extrapolating from these examples, we propose a hierarchical model of sSVs, in which individual 'blocks' display variations within themselves, but, additionally, sequence is prone to be exchanged between blocks with inversions as the transport media. While Strand-Seq data were unavailable in the sample carrying the 15q25 inversion, we orthogonally validated the SVs shown in SOTOS syndrome and 8p23.1, as well as providing further support for 22q11 (Fig. S35).

## Discussion

We present NAHRwhals, a long-read assembly-based workflow for identifying and deconvoluting regions of overlapping NAHR-mediated rearrangements. NAHRwhals works by determining local alignments of a region of interest to a reference, and subsequent segmentation of the alignment to summarize syntenic blocks via 'compressed dotplots'. This massive reduction in data complexity allows us to apply an exhaustive search strategy in which we effectively test all possible sequences of NAHR events up to depth 3 (or higher; at the cost of computation time). Thus, our method closes an important gap in current methodology: to enable the automatic detection and study of genome complexity caused by overlapping and consecutive repeat-mediated rearrangements. NAHRwhals can be applied to different mammalian species, as highlighted through great ape and tomato based analyses, enabling to further the understanding of the impact of complex rearrangement processes during genome evolution. SVs associated with segmental duplications are not yet routinely detected by most SV callers, and, likewise, genomic rearrangements with multiple breakpoints are still difficult to spot algorithmically. With NAHRwhals, we actively embrace SDs and multiple breakpoints, and demonstrate that the respective complex rearrangements under study follow rules that can be modeled. NAHRwhals and its sSV hypothesis can thus lead to new SV discoveries (e.g. three Mbp-sized SVs in T2T) and insights into these complex regions that might have implications in disease or other phenotypes as well as evolution.

Our representation of alignments in 'segmented' dotplots bears parallels to genome assembly or pan genome graphs, with NAHRwhals segments being analogous to nodes in a graph[59,60]. Following this analogy, the relative position of a NAHRwhals segments could be re-interpreted as links between nodes, with e.g. diagonally adjacent segments being considered 'linked'. In this representation, we reason that NAHRwhals SV calling could eventually be re-formulated as a graph traversal problem, which we consider a promising alternative to the currently employed brute-force algorithm.

We showcase the advances made by NAHRwhals by screening across 23 previously published[61] and 5 previously unpublished haplotype-resolved human genomic assemblies, revealing NAHR-mediated complexities in 37 loci and suggesting that these patterns are far more common in healthy individuals than current genomic studies have suggested[9]. Among all inferred sSV configurations, roughly two-thirds of predicted intermediate states could indeed be observed in other samples, supporting the notion that the majority of

such complexities have formed via serial accumulation of overlapping simple SVs, rather than by individual complex events.

As a currently remaining limitation, it should be noted that NAHRwhals does not incorporate information about population-based haplotype structures in its analysis. Indeed, population-based analysis of the sSV loci revealed highly complex inheritance patterns likely to be mediated by high rates of SV recurrence such as inversion toggling. As a result, actual locus histories could be more complex than what is predicted by the tool unless in the case of closely related individuals commonly found e.g. in patient trios. In evolutionary studies, we recommend validating the predicted sSVs using complementary methods to gain a more complete understanding of the locus history. Moreover, individual non-NAHR based complex rearrangement events can lead to highly complex patterns of variation, too[62], and these may in some instances be hard to distinguish from sSVs, especially in cases where intermediate haplotype structures are missing. The 'missing' intermediates were, without exception, associated with sSVs of allele count below three, suggesting that they may be missing from our callset due to the low sample number. Yet, the presence of flanking segmental duplications at the boundaries of the SV events studied here support a major role of NAHR even in these cases.

When extending our search to greater evolutionary distances, we note that NAHR alone is rarely sufficient to explain variation in sSV loci across four great ape assemblies. Assembly gaps did not contribute significantly to these unexplained variants, supporting the notion that over evolutionary timespans (mya), human sSV loci have seen large-scale rearrangement processes mediated by mechanisms distinct from NAHR. We note that in the future, greater numbers of high-quality ape assemblies will help paint a finer picture, especially concerning the phenomena of incomplete lineage sorting and recurrence[44].

Human NAHR-mediated complexities are strongly (3 to 6-fold) enriched in regions containing recurrent inversions and disease associated CNVs and can explain variation in medically relevant genes such as *TPSAB1*. sSVs in particular show a dynamic interplay with sequences known to be at risk for CNVs. Such interactions are not unexpected given that morbid CNV regions are frequently flanked by, and nested in, complex repeat patterns which mediate their formation[14,51,52,63]. We identify at least two modes of interaction. Firstly, the individual flanks of such regions are prone to harboring sSVs, altering the SD content available for formation of the 'main' CNV. In other cases, we also observe 'hierarchical' events in which large inversions span regions of CNV risk and contribute to exchanging sequence between the flanks, putatively increasing their complexity over time, or creating new hotspots of interspersed SDs which can again diversify over time.

The choice of the reference sequence is important as alleles might change[64]. We showcased this by comparing GRCh38 and CHM13-T2T, where the former suffered from unresolved sequences. These two references frequently represented alternative versions of alleles, some of which are likely attributable to falsely collapsed or duplicated sequences in GRCh38[65], whereas in other cases they may be attributable to repeated mutation and high variability. As highlighted, the dynamic nature of these sequences is especially important as many segmental duplications have an impact on medically relevant genes or other important phenotypes[22] and thus motivate a close analysis also in other organisms. To accomplish this, NAHRwhals can be adjusted to searches across species or even outside of hominidae. Due to the default resolution of 1 kbp for segment compression, repeats shorter than 1 kbp are likely to be missed by our algorithm and are thus not modeled as substrates for NAHR. Notable candidate for short SV-mediating repeats are e.g. pairs of Alu-elements, although their mode of formation likely involves mechanisms other than NAHR, and they mediate predominantly short SVs of few 100 bps to <5 kbp[66]. Thus, while sSV-like mechanisms might be observable also at such smaller scales, due to their size we do not expect such SVs to play a major role in the predisposition of the sSVs described here.

The maximum depth (default: 3) during the breadth-first exhaustive search for mutations can in principle also be a limiting factor in finding correct solutions. Although we have not encountered loci exhibiting depth-four SVs, we have left the option for higher depth exploration open to the user, and also exposed heuristics to control the otherwise exponential growth of nodes to explore at higher depths. However, four or more consecutive SVs would also indicate that a locus has undergone such extreme levels of restructuring that these SVs likely require more detailed analysis, and we therefore do not consider the depth a highly limiting factor.

A natural future extension of NAHRwhals would be also to take into account the percentual identity of segments to improve alignment scoring, rather than considering all segments as 'perfect' alignments. We expect that this approach would help selecting the correct optimal solution at the end of the exhaustive mutation search. At the same time, however, even such an approach will suffer from the often prohibitively complex variation patterns in SDs owing to inversion recurrence, sequential cycles of duplication and deletion, and frequent gene conversion events[14,22].

For obvious reasons, NAHRwhals is also reliant on the correctness of the assembly itself. For this study, we have assessed the correctness of most assemblies by realigning ultra long ONT reads to themselves and screening for homozygous SV across them. Such SV would indicate assembly errors such as rearrangements that would impact the results of NAHRwhals. Over the screened assemblies, no mis-arrangements were observed, and indels indicative of collapses or duplications were rare, too (~5% of sequences) (Fig. S33).

Another important point to accomplish is the automatic parameter tuning that is happening in NAHRwhals. Firstly, we observe a general robustness of the dotplot encryption. Still, our level of compression scales with overall sequence length, meaning that SV predictions can be sensitive to the size of the region of interest – i.e., SVs that affect only a small portion of the window (<10%) can be missed in some cases. Furthermore, by default we consider sequence variation as 'explained' if mutated alignments show more than 98% congruence, again discarding variants much smaller than the sequence window. It is likely that a portion of sSVs adjacent to very long CNV sites may have been missed by our survey in this way. Thus, local rearrangements may appear to be unexplained from a narrow angle, but may be explained when the broader sequence context is taken into account.

Overall, this work demonstrates the significance of complex NAHR-shaped variation and its ubiquitous detection across newly assembled genomes, regardless of species. These regions have been demonstrated to be of critical importance in various phenotypes, including those related to disease. NAHRwhals allows for the automatic detection and study of these regions across multiple assemblies, as well as the identification of the events that likely lead to the complex patterns that are currently observed. With the rapidly increasing number and quality of novel genome assemblies, we anticipate that NAHRwhals will be instrumental in uncovering the origins of disease-causing variants in patients and advancing our understanding of the evolution of these highly variable regions of the genome.

## Methods

### Pairwise sequence alignments

To obtain accurate pairwise alignments even in highly repetitive genomic regions, a custom pipeline was built around the minimap2 aligner (version 2.18)[32] to create pairwise alignments: Before aligning, the *Query* sequence is split into chunks of 1 kbp by default. The 'chunks' are then aligned to the *Ref* sequence separately (using the minimap2 parameters *-x asm20 -P -c -s 0 -M 0.2*; see Fig. S2), reducing the need for read-splitting, which is known to be error-prone in minimap2[11]. The choice of chunklength represents a tradeoff, as (a) too small chunks tend to produce overly fragmented alignments which lead to long computation time and a tendency for shorter alignments, and (b) overly large chunks tend to ignore or over-merge short alignments. In practice, alignments have proved relatively robust towards the choice of chunklength (Figs. S36, S37), justifying our default choice. In a post-processing step, alignment pairs are concatenated whenever the endpoint of one alignment falls in close proximity to the startpoint of another (base pair distance cutoff: 5% of the chunk length). If multiple alignments 'compete' for the same partner (e.g., two alignments ending close to the beginning of another), only the longest 'competitor' alignment gets selected for merging.

### Noise-reduction in pairwise alignments for subsequent segmentation

Pairwise alignments are retrieved from minimap2 in *.paf* format, in which each alignment can be interpreted as a four-dimensional vector from start (query-start/target-start) to end (query-end/target-end) coordinates. To prepare subsequent compression steps, alignments are pre-processed in multiple ways: First, alignments are filtered by a minimum length threshold ($l$), removing very short alignments. Second, alignment breakpoint coordinates are rounded in x and y direction to the closest multiple of a rounding parameter ($r$). Finally, alignment vectors are shortened along the x or y axis in case they do not have a slope of exactly 1 or -1 until they do so. The choice of $r$ and $l$ directly influence the expected dimensionality and complexity of the segmented dotplot. To maximize information content while limiting the size of segmented dotplots, we set l and r to be stepwise functions of the sequence length:

(1)  $l = r = 100$ for sequences smaller than 50 kbp
(2)  $l = r = 1.000$ for sequences between 50 kbp and 500 kbp
(3)  $l = r = 10.000$ for sequences between 500 kbp and 5 Mbp
(4)  $l = r = 20.000$ for sequences larger or equal to 5 Mbp

### Alignment segmentation and dotplot condensation

A custom algorithm converts the pairwise alignment returned by minimap2 into a *segmented* representation (NAHRwhals step 3) which facilitates downstream analysis and implicitly identifies repeat pairs required for subsequent steps. Following noise-reduction, borders, or *'gridlines'*, separating unique sequence blocks, are inferred in an iterative way (Fig. S3). In the first iteration, horizontal and vertical gridlines are drawn starting from each start- or endpoint of any alignment. In every subsequent step, overlaps between existing gridlines and alignments are determined, with the points of overlaps serving as a new source for spawning a new *gridline* in perpendicular direction. This process is repeated until no new gridlines can be spawned. In rare cases where the determined grid exceeds predefined maximum dimensions (default: n_rows + n_cols <= 250; n_alignments <= 150), the parameters for minimum alignment length (l) and rounding (r) are doubled until dimension requirements are met. Any sequence interval between two gridlines is then considered a *segment*, and any pairwise alignment can be expressed as a matrix of alignment of segments (x: Ref, y: Query) (Fig. S3B).

### Exhaustive search for sSVs

The information reduction obtained from condensing potentially multi-Mb alignments to a segmented matrix of much smaller dimensions allows us to employ an exhaustive search strategy to identify serial SVs capable of transforming the *Ref*-configuration into that of *Query*. We focus on exploring NAHR-mediated SVs (deletions, duplications and inversions), and a key observation is that segmented dotplots retain information of repetitive sequences (i.e. rows or columns with >1 colored square). Starting from one segmented dotplot, a recursive *breadth-first* algorithm initially identifies repeat pairs in *Ref* (pairs of colored squares in the same row) and infers possible NAHR-mediated SVs (Del/Dup between similarly colored squares; Inv

between opposites). Subsequently, every SV is being simulated by deleting, duplicating or inverting respective columns of the segmented dotplot, and new higher-depth SVs are inferred and simulated as the search progresses. After each simulated mutation, the similarity between *Query* and the mutated *Ref* is determined (see "Methods" section *"Scoring of mutated segmented matrices"*). We implemented the following heuristics (i) Limit the maximum allowed number of duplications of any segment to avoid 'exploding' sequence length [default: 2]. If any mutated matrix has reached the maximum number of duplications, it will not be further duplicated. (ii) Per layer, retain only the X best-scoring nodes [default: all nodes]. (iii) Limit the maximum depth of the search [default: 3]. Finally, we report all optimal (score = 1) SV trajectories, as well as suboptimal SV trajectories within 5% of the score of the best trajectory.

## Scoring of mutated segmented matrices

Mutated matrices are assigned a percent alignment score using a linear-time heuristic greedy alignment. Positively aligning segments are treated as 'matches', everything else is treated as a mismatch. Starting at the first position of the query sequence, we identify the closest match in the reference and assign it a score according to the following recurrence, where i runs over min(query_positions, reference_positions) and j runs over positions in the reference:

```
score[i] = max j. (if is_match(i,j) then 1 / (1 + |offset[i]
+ 1 - j|) else 0)
offset[0] = 0
offset[i+1] = if not (exists j. is_match(i,j))
              then offset[i] + 1
              else argmin j. (if is_match(i,j) then |off
              set_i+1-j| else ∞)
final_score = (sum i. score[i]) / (reference_length +
query_length - offset[end])
```

If we restrict the maximum gap size to consider |offset_i + 1-j|, this recurrence runs in linear time. Additionally, the score equals 1 only for a perfect alignment and decreases with the number of gaps and mismatches, making it a useful heuristic for our SV trajectory search.

## Whole-genome mode

NAHRwhals can also be run in whole-genome mode, which automates the identification of Regions of Interest (ROIs), i.e. potentially SV-carrying regions, to be scanned subsequently. Conceptually, ROIs are derived from alignment breaks in a whole-genome alignment of Ref and Query. In detail, given two assembly fasta files, Ref (e.g., hg38) and Query (e.g., a single-haplotype de novo assembly), we

1) run a whole-genome alignment with minimap preset '-x asm5'.

2) Remove alignments which are entirely contained within other alignments in Ref and Query coordinates.

3) Extract Ref and Query coordinates of all alignment breakpoints.

4) Exclude pairs of breakpoints closer than 10kbp in *Ref* and *Query* coordinates, which are typically indicative of smaller insertions and deletions.

5) Merge breakpoints to create windows of interest. We repeat this process with three different merge distances; 200 kbp, 1 Mbp, 5 Mbp to account for differently sized SVs, creating partially overlapping windows.

6) Expand each window by 50% in each direction.

7) Remove windows <50kbp after expansion; the resulting list are the ROIs.

8) Invoke NAHRwhals SV calling on each window.

9) Post-processing: Should an SV call between two partially overlapping windows be discordant, keep the call made by the smallest ('most zoomed-in') window.

## Simulation experiments

For each pair of "SD similarity" (90%, 95%, 99%) and "SD length" (100 bp, 500 bp, 1.000 bp, 10.000 bp), we created 50 genomic sequences with 2 pairs of non-overlapping SDs of randomized position and orientation each. Subsequently, we simulated a pool of sequence derivatives, realizing two randomly chosen NAHR-concordant combinations of Inv, Del and Dup up to depth 3 each, totaling 1200 mutated sequences (see Data availability). These mutated sequences and their unmutated 'ancestor' were given to NAHR-whals for SV calling and results were compared with the known background of sequences.

## Collection of 336 sSV-regions of interest

In order to maximize the scope of our survey, we based our sSV search on a set of all ($n$ = 107,590) SV regions from a previous large-scale SV survey of 64 human haplotypes[9], and additionally considered polymorphic inversion calls ($n$ = 399), many of which are known to be associated with complex variation[14]. Given that individual SV calls may sometimes be part of the same sSV block, we devised a specialized strategy to merge individual, spatially nearby SVs into broader 'SV;Repeat-containing' regions using the following procedure: (1) filter variants to length >10 kbp to exclude the bulk of non-NAHR events. (2) merge SVs with any overlapping segmental duplications (to include mutation-mediating SDs in the loci). (3) Merge any intervals if they have at least 50% overlap. (4) Elongate each region by 25% of its length on either end. (5) Subtract centromeres and ALR repeat regions. (6) Merge regions with <100 kbp distance to each other (7) filter to regions >40 kbp. The resulting regions contain between one and several thousand SV/Inv calls, with region lengths ranging from 40 kbp to 26 Mbp (median: 170.8 kbp). The scripts and all input data are available at https://github.com/WHops/NAHRwhals_rois.

## Collection of 48 morbid CNV-regions of interest

Loci containing disease-associated CNVs were based on four separate lists from Cooper et al.[50] (Table 1 in Cooper et al; 44 regions. Table 2 in Cooper et al: 14 regions) and[56] (Table S2 in Cooper et al.; 19 regions. Table S3 in Cooper et al: 36 regions), totaling at 113 loci. All regions were transformed from hg18- to hg38 coordinates using UCSC liftover[67]. Next, we applied the same 7-step merging strategy as used for defining 336 SV-Inv-sSV regions (see methods:Collection of 336 sSV-regions of interest), resulting in 48 nonredundant regions of interest which were subsequently used for analysis.

## Human and great ape assembled haplotypes

We based our analysis on 56 human and four great-ape assemblies. The human haplotypes consisted of the GRCh38 and CHM13-T2T assemblies and 56 de novo assemblies based on PacBio HiFi reads produced by the HGSVC consortium (see Data availability) as previously described (Ebert et al)[9]. Samples considered for assemblies were of diverse ancestry, including individuals from five major superpopulations (African: 16 individuals, Ad Mixed American: 3, European: 4, South Asian: 1, South East Asian: 4), Phased genome assemblies were created in two batches with slightly different procedures:

The first batch (14 / 28 samples, HG00512, HG00513, HG00514, HG00731, HG00732, HG00733, HG02818, HG03125, HG03486, NA12878, NA19238, NA19239, NA19240, NA24385) was produced by an improved version of the PGAS pipeline[9,68] (PGASv13). Briefly, a non-haplotype resolved assembly was created with hifiasm v0.15.2[69] after removing potential adaptor contamination in the HiFi reads. This draft assembly was then used as reference in subsequent steps employing Strand-seq data to cluster assembled contigs by chromosomes and to create a phased set of genomic variants during the so-called integrative phasing step of PGAS. The phased set of variants then informed the haplotagging of the HiFi reads, leading to two haplotype-specific read

sets per sample. The final phased assemblies were then created by running hifiasm on each haplotype-specific read set. Basic characteristics of the phased assemblies were reported in previous work (see Ebler et al. [61], Supplementary Data 1).

Assemblies of the remaining 14 samples (GM19129, GM19434, HG00171, HG00864, HG02018, HG02282, HG02769, HG02953, HG03452, HG03520, NA12329, NA19036, NA19983, NA20847) were produced by hifiasm v0.16.1-r375[69] using PacBio Hifi data of 27.7–47.2X coverage. For 4 samples with parental short read sequencing data[40], we used the trio binning assembly mode of hifiasm (GM19129, HG02018, NA12329, NA19983) and for 6 samples with paired-end short read Hi-C data[70], we used the Hi-C phasing mode of hifiasm (GM19434, HG02282, HG02769, HG02953, HG03452, HG03520). For the remaining 4 samples, hifiasm was run without additional data types (HG00171, HG00864, NA19036, NA20847). Six of the samples (GM19129, HG02282, HG02769, HG02953, HG03452, HG03520) have not been reported on by the HGSVC previously. The average CPU time of hifiasm was 323 h at a peak memory usage of 105 Gb.

Additionally, four hifiasm-based great-ape assemblies (Bonobo, Chimpanzee, Gorilla and Orangutan) were acquired from a recent publication[22] (available at https://doi.org/10.5281/zenodo.4721957). In these four datasets, contigs labeled as 'primary' were used.

### Core-duplicons

sSV loci were tested for overlap with genes and gene families mapping to 'core-duplicons'[53]. To create a table of relevant genes, we assembled a list of 22 prominent core duplicon - associated gene families (*NBPF, RANBP2, RGPD, PMS2, PPY, C9orf36, ZNF790, SPRYD5, NPIP* (also known as *Morpheus*), *GOLGA, LRRC37, TBC1D3, USP6, SMN, CCDC127, TRIM51, GUSBP, FAM75A, SPATA31, OR7E, DPY19, SPYDE*)[49,71]. We subsequently queried the gencode gene annotation (version 35)[72] for all members of those families, leading to a list of 131 gene instances (Supplementary Data 3).

### Sequence validation using Strand-Seq

Strand-Seq is a single-cell DNA sequencing technique which allows to infer the directionality of genomic sequence based on the directionality of strand-specific short-reads mapping to the genome alone[35]. Strand-Seq based genotypes were created using the ArbiGent module of Mosaicatcher, a software for the analysis of Strand-Seq data[14,73]. ArbiGent requires a minimum of 500 bp of sequence which can be uniquely mapped with short reads, which was the case for 11/37 sSV regions for which inversion genotypes were obtained. In two regions, 'stage-2'-deletions and duplications were mappable with Strandseq (chr1:119760829-121619186 and chr1:108185627-108570960), providing exact Strand-Seq genotypes for these loci. In the remaining nine regions, StrandSeq reported the presence/absence of inverted sequence, which could be cross-checked for consistency with NAHRwhals-based SVs (e.g., Nahrwhals: inv-del was considered to be consistent with Stand-Seq: inv). Visualizations of Strand-Seq data were made with a custom pipeline available at: https://github.com/WHops/sseq_plot. For one region, chr1:108185627 INV-DEL and INV-INV, we analyzed the fraction of reads aligning to the segmental duplications in forward or reverse orientation[74] (Fig. S24) as a proxy for the amount of SD sequence in forward or reverse orientation. For each sample, all reads mapping to the largest SD in the region (displayed in green in Fig. S24) in negative and in positive direction were counted, and the fraction of negative-to-positive calculated. To account for read sampling uncertainty due to the low coverage of Strand-seq data we calculated 95% confidence intervals for the real fraction of negative-to-positive reads using the binom.test function in R.

### Integration of NAHRwhals-based calls with external SNPs

As part of our analysis of population-based SNP patterns, we combined our SV callset with an external SNP callset from[40]. A custom approach around WhatsHap[41] was used to assign the NAHRwhals-based SVs to the corresponding haplotype in the external SNP callset. For each SV, the SV-carrying sequence was extracted from its assembly, cut into 10kbp-long, non-overlapping chunks and mapped to the hg38 reference. WhatsHap haplotag was then invoked to assign chunks to h1 or h2 based on the external SNP callset, and the SV was finally assigned to the haplotype to which the majority of chunks were assigned. The code for this procedure can be found in a github repository: https://github.com/WHops/nahrwhals_phasing.

### Automated sequence validation with Nanopore reads

We used ONT reads generated by the HGSV Consortium for 11/26 samples - GM19129, HG00512, HG00731, HG00733, HG02282, HG02769, HG02818, HG02953, HG03452, HG03520, NA19239 (data availability). After removing adapters with Porechop (https://github.com/rrwick/Porechop), ONT reads were mapped to both haplotypes of their respective samples (i.e., reads from each sample were mapped both to the h1 and h2 assembly). We invoked the sniffles2[38] SV caller to identify homozygous variants which correspond to assembly regions not supported by any reads, i.e. likely assembly errors. Using the lift-over coordinates provided by NAHRwhals, we queried the sequences corresponding to their CHM13-T2T-counterpart individually for each assembly.

### Validation of four morbid CNV-associated regions with Nanopore reads

Four disease-associated regions were additionally verified using a manual approach. For this, aligned ONT reads were phased post-hoc using 'samtools phase' and visualized in IGV, highlighting discordant reads which can be indicative of large-scale rearrangements (Fig. S19).

### Reporting summary

Further information on research design is available in the Nature Portfolio Reporting Summary linked to this article.

## Data availability

PacBio HiFi sequencing data, Strand-seq as well as Oxford Nanopore sequencing data were generated by the HGSVC consortium and can be accessed through the HGSVC data portal https://www.internationalgenome.org/data-portal/data-collection/structural-variation. Phased SNPs used for the population-based SV analysis are available at https://www.internationalgenome.org/data-portal/data-collection/30x-grch38. Assembled genomes can be accessed in the zenodo database via https://doi.org/10.5281/zenodo.7635935. High-resolution views of Supplementary Figs. S15-S22 (37 human sSV loci) and Supplementary Figs. S30-S32 (sSV loci in great ape genomes) have been deposited at https://doi.org/10.5281/zenodo.13107026. Great ape genome assemblies (Chimpanzee, Bonobo, Gorilla, Orangutan) have been taken from a recent publication[22] who have made them available at https://doi.org/10.5281/zenodo.4721957. The set of all (n = 107,590) SV regions from a previous large-scale SV survey of 64 human haplotypes (Ebert et al.)[9] is available at https://ftp.1000genomes.ebi.ac.uk/vol1/ftp/data_collections/HGSVC2/release/v1.0/integrated_callset/freeze3.sv.alt.vcf.gz. 1200 artificially created sequences carrying NAHR-based mutations used for benchmarking are available under the following https://doi.org/10.5281/zenodo.13363005. Supplementary Data 1–4 are provided as supplementary files and are also available under https://doi.org/10.5281/zenodo.13363333.

## Code availability

NAHRwhals is available on github under an MIT license: https://github.com/WHops/NAHRwhals. Code for SV phasing and integration with external SNPs (population-based analysis) is available at

https://github.com/WHops/nahrwhals_phasing. Visualizations of Strand-Seq data were made with a custom pipeline available at: https://github.com/WHops/sseq_plot. Code used to create artificial sequences and their mutated counterparts for simulation-based benchmarking is available at https://github.com/WHops/nahrwhals_simulate_events.

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

## Acknowledgements

Funding was provided by: National Institutes of Health (NIH) grant U24HG007497 (to J.O.K.), 1U01HG011758-01 (to F.J.S), Federal Ministry of Education and Research (BMBF) grant 031L0181A (LAMarCK to J.O.K). The EMBL International PhD Program provided additional support. We thank members of the HGSVC for allowing us to incorporate several previously unpublished whole genome assemblies, along with raw data in the form of ultralong DNA reads, in this study. We thank Celia Tsapalou and Thomas Weber for valuable feedback on the code and the manuscript, and the members behind the deutsch_studio account on fiverr.com for designing our logo.

## Author contributions

Conceptualization and Methodology were envisioned by W.H., J.O.K and F.S. Development of the Software was performed by W.H and M.J. and subsequent formal analysis by W.H. with supervision by F.S and J.O.K. Phased human genome assemblies from long-read genomes generated by the Human Genome Structural Variation Consortium and T.R. Writing was performed jointly by W.H, J.O.K and F.S with input from all authors.

## Funding

## Competing interests

J.O.K. has previously disclosed a patent application (no. EP19169090) relevant to Strand-seq. F.J.S receives research support from Illumina, PacBio and ONT. The other authors declare no competing interests.

## Additional information

## Human Genome Structural Variation Consortium (HGSVC)

Hufsah Ashraf[6], Peter A. Audano[7], Ola Austine[3], Anna O. Basile[8], Christine R. Beck[7], Marc Jan Bonder[9], Marta Byrska-Bishop[8], Mark J. P. Chaisson[10], Zechen Chong[11], André Corvelo[8], Scott E. Devine[12], Peter Ebert[13], Jana Ebler[6], Evan E. Eichler[14], Mark B. Gerstein[15], Pille Hallast[7], William T. Harvey[14], Patrick Hasenfeld[1], Alex R. Hastie[16], Mir Henglin[6], Kendra Hoekzema[14], Wolfram Höps[1], PingHsun Hsieh[14], Sarah Hunt[3], Miriam K. Konkel[17], Jan O. Korbel ®[1,3,24]✉, Jennifer Kordosky[14], Peter M. Lansdorp[18], Charles Lee[7], Wan-Ping Lee[19], Alexandra P. Lewis[14], Chong Li[20], Jiadong Lin[21], Mark Loftus[17], Glennis A. Logsdon[14], Tobias Marschall[6], Ryan E. Mills[22], Yulia Mostovoy[23], Katherine M. Munson[14], Giuseppe Narzisi[8], Andy Pang[16], David Porubsky[14], Timofey Prodanov[6], Tobias Rausch ®[1,2], Bernardo Rodriguez-Martin[1], Xinghua Shi[20], Likhitha Surapaneni[3], Michael E. Talkowski[23], Feyza Yilmaz[7], DongAhn Yoo[14], Weichen Zhou[22] & Michael C. Zody[8]

[6]Institute for Medical Biometry and Bioinformatics, Medical Faculty, Heinrich Heine University, Düsseldorf, Germany. [7]The Jackson Laboratory for Genomic Medicine, Farmington, CT, USA. [8]New York Genome Center, New York, NY, USA. [9]Division of Computational Genomics and Systems Genetics, German Cancer Research Center (DKFZ), Heidelberg, Germany. [10]Department of Quantitative and Computational Biology (QCB), University of Southern California, Los Angeles, CA, USA. [11]Department of Biomedical Informatics and Data Science, Heersink School of Medicine, University of Alabama, Birmingham, AL, USA. [12]Institute for Genome Sciences, University of Maryland School of Medicine, Baltimore, MD, USA. [13]Core Unit Bioinformatics, Medical Faculty and University Hospital Düsseldorf, Heinrich Heine University, Düsseldorf, Germany. [14]Department of Genome Sciences, University of Washington School of Medicine, Seattle, WA, USA. [15]Computational Biology and Bioinformatics Program, Yale University Medical School, New Haven, CT, USA. [16]Bionano Genomics, San Diego, CA, USA. [17]Department of Genetics & Biochemistry, Clemson University, Clemson, SC, USA. [18]Terry Fox Laboratory, BC Cancer Agency, Vancouver, BC, Canada. [19]Department of Pathology and Laboratory Medicine, Perelman School of Medicine, University of Pennsylvania, Philadelphia, PA, USA. [20]Department of Computer and Information Sciences, Temple University, Philadelphia, PA, USA. [21]Xi'an Jiaotong University, Xi'an, China. [22]University of Michigan Medical School, Department of Computational Medicine and Bioinformatics, Ann Arbor, MI, USA. [23]Cardiovascular Research Institute and Institute for Human Genetics, UCSF School of Medicine, San Francisco, CA, USA.

