## [Peer Review File · Nature Communications]

Impact and characterization of serial structural variations across humans and great apesEditorial Note: This manuscript has been previously reviewed at another journal that is not operating a transparent peer review scheme. This document only contains reviewer comments and rebuttal letters for versions considered at *Nature Communications*.

REVIEWERS' COMMENTS

Reviewer #1 (Remarks to the Author):

The authors tackle the fascinating and challenging problem of inferring and reconstructing serial SVs which are expected to be enriched in regions harboring segmental duplications. My overall assessment of the work is that it is novel, creative and would support its publication. It is almost certainly not the last word in this space, but is an impressive and interesting start.

The manuscript has already undergone thorough consideration by two reviewers. The authors have generally responded appropriately and thoroughly to all reviewer comments and requests. I have one area where I would like to request additional attention.

Like Reviewer 1, I found the description of the methodology difficult to follow (Reviewer 1.Minor.7). It is improved in the new version with the more detailed descriptions, but a combination of jargon, inconsistent word usage, and unclear use of "/" , reduce the clarity of the writing. A non-exhaustive initial list of suggestions to improve writing follows:

1. "Given reference coordinates of interest and reference/assembly fasta files, Minimap2 32 is employed to find the region of interest in the assembly fasta. The region of interest is then extracted from both fasta files for further analysis."

This would be much clearer if the authors could initially define and use throughout terms like query and target, if possible, instead of the ambiguous "reference/assembly". Minimally defining and using reference and assembly separately would be better.

2. Step iv) is summarized as 'simulating chains of SVs' and subsequently 'Exhaustive mutation search'. A minor point but this is inconsistent and reduces clarity.

3. I would recommend considering re-writing all uses of "/" for clarity.

4. I would also recommend moving

"As input, NAHRwhals requires two assembly fasta files (the first one used as 'reference') and coordinates of a reference locus of interest."

to just before i) Sequence retrieval, then adapting the text that follows.

Reviewer #1 (Remarks on code availability):

I reviewed the code and documentation, but was unable to test it as I could not install it on my computer in a timely manner. Documentation is quite terse, but detailed enough that it should be possible to run.

Reviewer #2 (Remarks to the Author):

The authors comprehensively addressed the comments of the reviewer.

There are still some typos:

- * page 3: "providednew"

- * page 12: an occurrence of "bonobo and orangutan" are not capitalized

- * page 15: "that can, under the sSV hypothesis, would be"

* page 21: "iterative way. (Figure S2)." (remove the first .)

Reviewer #2 (Remarks on code availability):

Repository does not contain any example data. It is possible to run the tests in R but in that case it is not clear what are the input data. I have not been able to open the output pdf (in res/chr1_partial-1700000-3300000/Fasta_ref_Fasta_asm_all.pdf) since it was corrupted. The pdf/png files in res/chr1_partial-1700000-3300000/self/pdf are correct.

NAHRWHALS REVIEWS - ROUND 2 (July/August 2024)

We thank both reviewers sincerely for their help in improving our manuscript and recognizing the importance of our work. We indicate in blue our replies to the last remaining questions.

Reviewer #1 (Remarks to the Author):

The authors tackle the fascinating and challenging problem of inferring and reconstructing serial SVs which are expected to be enriched in regions harboring segmental duplications. My overall assessment of the work is that it is novel, creative and would support its publication. It is almost certainly not the last word in this space, but is an impressive and interesting start.

The manuscript has already undergone thorough consideration by two reviewers. The authors have generally responded appropriately and thoroughly to all reviewer comments and requests. I have one area where I would like to request additional attention.

Reviewer 1. 1 Like Reviewer 1, I found the description of the methodology difficult to follow (Reviewer 1.Minor.7). It is improved in the new version with the more detailed descriptions, but a combination of jargon, inconsistent word usage, and unclear use of “/”, reduce the clarity of the writing. A non-exhaustive initial list of suggestions to improve writing follows:

1. “Given reference coordinates of interest and reference/assembly fasta files, Minimap2 32 is employed to find the region of interest in the assembly fasta. The region of interest is then extracted from both fasta files for further analysis.”

This would be much clearer if the authors could initially define and use throughout terms like query and target, if possible, instead of the ambiguous “reference/assembly”. Minimally defining and using reference and assembly separately would be better.

2. Step iv) is summarized as ‘simulating chains of SVs’ and subsequently ‘Exhaustive mutation search’. A minor point but this is inconsistent and reduces clarity.

3. I would recommend considering re-writing all uses of “/” for clarity.

4. I would also recommend moving

“As input, NAHRwhals requires two assembly fasta files (the first one used as ‘reference’) and coordinates of a reference locus of interest.”

to just before i) Sequence retrieval, then adapting the text that follows.

Response: In light of the repeated highlight particularly of this section by both reviewers, we have completely re-written the technical summary of NAHRwhals, taking into account all suggestions above. We edited the Methods section accordingly to ensure consistent use of terminology.

We believe that the section has now gained in clarity. The new text reads as follows:

Automated detection of serial structural variations (sSV) from genome assemblies.

We developed the NAHRwhals framework to allow systematic identification of sSVs in haplotype-resolved genome assemblies. NAHRwhals can be run in two primary modes: genotyping mode and whole-genome mode. The required inputs are:

- (1) Reference Genome ('*Ref*'): a reference FASTA file, such as GRCh38.
- (2) Query Genome ('*Query*'): A single-haplotype assembly FASTA file to be analyzed.
- (3) Regions of Interest ('*ROI*'s) - Required only in genotyping mode: coordinates on the reference genome ('*Ref*') to be genotyped. In whole-genome mode, NAHRwhals automatically determines *ROIs* by conducting an initial alignment of the entire reference and query genomes to identify discordant regions, which are then used as *ROIs* (Methods, Figure S1).

NAHRwhals genotypes any *ROI* in four steps: i) isolating the *ROI* in *Ref* and locating its counterpart in *Query*, ii) aligning the *ROI*-sequences from *Ref* and *Query*, iii) turning this pairwise alignment into a simplified, '*segmented*', representation which facilitates repeat discovery and SV simulation, and iv) employing a depth-first search to find sSV candidates that can rearrange the segments in *Ref* to mimic their order in *Query* (see also Figure 1B; Methods). Below, we describe these four steps in detail:

i) *Sequence retrieval*: The *ROI* is extracted from the reference genome (*Ref*), resulting in *ROI-Ref*. Minimap2³² is then used to locate the corresponding, potentially SV-carrying, region in the query genome (*Query*), yielding *ROI-Query*.

ii) *Highly accurate pairwise alignments*: A pairwise alignment between *ROI-Ref* and *ROI-Query* is produced using a custom pipeline. This pipeline involves splitting *ROI-Query* into chunks of 1 kbp, aligning these chunks to *ROI-Ref* individually (allowing for multi-mappings), and subsequently re-joining them (Methods, Figure S2). This method significantly improves alignment quality in repeat-rich regions compared to the default Minimap2 settings (Methods). Alignments shorter than 500 bp are then discarded, and alignment coordinates are rounded to the nearest multiple of a rounding factor (default: 1 kbp for $ROI \leq 500$ kbp, 10 kbp for $ROI > 500$ kbp) to eliminate small alignment incongruencies.

iii) *Alignment Segmentation*: A segmentation algorithm simplifies the alignment by identifying uninterrupted stretches of alignment, referred to as "segments," which range in size from 1 kbp to several hundred kbp (Methods, Figure S3). This segmented representation of the alignment retains all information of the alignment, while significantly simplifying the identification of NAHR-enabling repeat pairs and the exhaustive search for sSVs in the subsequent step.

iv) *Exhaustive search for sSVs*: The segmented alignment, represented as a matrix, serves as the foundation for an exhaustive search for NAHR-based serial structural variations (sSVs). This search employs a breadth-first search approach to explore possible trajectories of serial SVs, with the goal of transforming the reference genome segments (*ROI-Ref*, represented on the x-axis) into the corresponding query genome structure (*ROI-Query*). At each stage of the search, the algorithm:

- (1) generates a list of potential downstream NAHR-based SVs (deletions and duplications between pairs of segments in the same orientation, and inversions between pairs in inverse orientation).
- (2) Simulates each SV by deleting, duplicating, or inverting respective columns of the alignment matrix, generating new pairwise alignment matrices.
- (3) Calculates a 'segmented alignment score' which quantifies the percentage of correctly aligned segments between *ROI-Ref* and *ROI-Query*, scaled by segment size (Methods).

The algorithm explores this space up to a predefined maximum depth [default: 3], using a tree structure where each node represents a state of the genomic locus after applying previous SVs. Serial SVs with a segmented alignment score within 5% of the best scoring one are reported. For downstream analyses, an sSV simulation is considered 'successful' if it achieves an alignment score above a given threshold [default: 98%]. As default heuristics, we limit the maximum allowed number of duplications per serial SV to avoid 'exploding' sequence length [default: 2] and, per layer, retain only the X best-scoring nodes [default: Inf].

Reviewer 1.2: Remarks on code availability

I reviewed the code and documentation, but was unable to test it as I could not install it on my computer in a timely manner. Documentation is quite terse, but detailed enough that it should be possible to run.

Response: We have tested the code on various UNIX systems where we found no installation problems. We observed that our installation instructions were not detailed enough for Apple Silicon (M1, M2, M3) machines, which require an additional configuration step for anaconda. We have added this additional info to the README. We have also improved the code documentation slightly fixed a bug mentioned by Reviewer #2 and added additional information on the implemented test case.

Reviewer #2 (Remarks to the Author):

The authors comprehensively addressed the comments of the reviewer.

Reviewer2.Minor1 There are still some typos:

- * page 3: "providednew"
- * page 12: an occurrence of "bonobo and orangutan" are not capitalized
- * page 15: "that can, under the sSV hypothesis, would be"
- * page 21: "iterative way. (Figure S2)." (remove the first .)

Response: We have fixed these typos in our manuscript.

Reviewer2.2:Remarks on code availability

Repository does not contain any example data. It is possible to run the tests in R but in that case it is not clear what are the input data. I have not been able to open the output pdf (in res/chr1_partial-1700000-3300000/Fasta_ref_Fasta_asm_all.pdf) since it was corrupted. The pdf/png files in res/chr1_partial-1700000-3300000/self/pdf are correct.

Response: The repository does indeed contain example data which are run in the tests in R, however they are somewhat hidden (inst/extdata) in compliance with R library standards. We have adapted the README so the user can more clearly identify which data are being used for the test. We have also identified the error the reviewer encountered regarding the corrupted pdf and fixed it.